# “New Balls Please”: Physical Load Imposed on Ball Boys during the Roland Garros 2022

**DOI:** 10.3390/ijerph20053793

**Published:** 2023-02-21

**Authors:** Cedric Brandli, Filip Svalina, Šime Veršić, Dario Novak

**Affiliations:** 1Research & Development, Holistic Tennis, 69340 Franchville, France; 2Faculty of Kinesiology, University of Zagreb, Horvaćanski Zavoj 15, 10000 Zagreb, Croatia; 3Faculty of Kinesiology, University of Split, Ulica Nikole Tesle 6, 21000 Split, Croatia

**Keywords:** ball kids, workload, tennis, youth athletic development, general health, speed

## Abstract

The process of becoming a ball kid at the French Open consists of different stages of selection and training. Selection and training of the ball kids is organized by the French Federation of Tennis (FFT) and is intended to be an immersive and educational experience. A sample was made up of ball kids participating at the 2022 French Open (Roland Garros). For this study, 26 ball kids were analyzed during several rotations of their activity on the court with different durations (N = 26; age = 15.00 ± 0.84; height = 169.03 ± 9.62; weight = 52.26 ± 7.35). Each ball kid participated in several analyzed rotations (data entry N = 94). Two groups are analyzed: ball kids at the net and in the back of the court. The result of the statistical analysis showed a statistically significant difference between the two groups in the variables: meters covered per minute on court (t = 6.85, *p* = 0.00), total number of decelerations per minute (t = 8.39, *p* = 0.00), walking and jogging meters per minute (t = 4.68, *p* = 0.00), and maximum velocity achieved (t = 3.02, *p* = 0.00). Participating as a ball kid during a professional tournament presents a unique experience for young athletes. Young people that are participating can improve their fitness, social skills, mental abilities, and well-being due to requests for the ball kids’ duties during match play and out of play activities.

## 1. Introduction

The ball kids of the French Open are recognized as the world’s best tennis ball kids and are often thanked by the players for their commitment and professionalism. The process of becoming a ball kid at the French Open consists of different stages of selection and training. Selection and training of the ball kids is organized by the French Federation of Tennis (FFT) and is intended to be an immersive and educational experience for all the kids. The kids who take part in the process are aged between 12 and 16 years old and are all members of the French Federation of Tennis, with various levels of play.

Every court on a professional tennis tournament has a trained squad of ball kids—boys and girls—who deal with game balls while not interfering with active play. Ball kids are individuals who retrieve and supply balls for players in professional tennis tournaments. Their activity helps speed up play and reduce inactive time. The ITF (International Tennis Federation) has approved procedures for ball kids that are produced by ITF Officiating [1].

There have been studies on the time-motion characteristics of young tennis players, including total distance and distance covered at different speeds [2,3], but to the best of the author’s knowledge, there have been no studies on the time-motion characteristics of ball kids. Time-motion characteristics in different game conditions include short-term, high-intensity movement, and using 10–15 Hz GPS to measure acceleration, deceleration, and the distance covered at different speeds gives better results than video-based analysis [4,5]. The same principle can be applied to the time-motion characteristics of ball kids because they have similar short-term, high-intensity movements between tennis rallies. Many studies are investigating external and internal loads in tennis matches. The development of technology tools like wearable and multivariable monitoring devices that include GPS features offers a practical way of monitoring those loads. External loads that are monitored that way are distances covered and acceleration and deceleration movements during tennis training and matches [2,3]. The rating of perceived exertion (RPE) scale is a way of measuring physical activity intensity [6]. This measure is considered a viable method for tracking internal loads using low-cost, accessible procedures [7]. 

Workload parameters for ball kids need to be tracked and analyzed to better understand the demands of the ball kid’s activity. Knowing how much time they spend on the court, how many meters they cover, and at what speed they perform is important for future selection and training processes. To better organize the ball kids by positions, it is vital to understand the difference in workload between positions on court. In this regard, this is the first study to quantify the physical demands placed on young children during the GS competition, allowing for appropriate load monitoring and the management of the risk of injury as well as the improvement of ball kids’ performance.

## 2. Materials and Methods

### 2.1. Participants

Ideally, the number of ball kids on the court should be six. They are positioned on different parts of the court: two ball kids are at the back of the left end of the court (one in each corner), two are at the back right end of the court, and two are at the net as shown in Figure 1. Ball kids should be scheduled so that they have sufficient rest during matches. Ball kids communicate and listen to the Chair Umpire’s instructions, and they are not on-court officials. Ball kids at the back of the court on the server’s side in between points hold their arms high above their heads, ready to pass balls if required or to indicate that they do not have a ball ready. The ball kids at the receiver’s end who have balls should roll them to one of the ball kids at the net between points. Ball kids at the net should crouch down, and if a ball is out of play and on the court, they should pick it up and return to the nearest position. At the end of every point, the ball kids at the net roll any balls they have to the server’s end of the court [1]. There are six ball kids on court during a match, and they are placed in two positions on court: at the back of the court and at the net.

For the 2022 edition of the French Open, 2500 kids registered to participate in the selection process for becoming one of the ball kids at the tournament. Selection lasted fourteen days and consisted of various playful activities that tested their social skills (team spirit and cooperation) and motor skills like agility, coordination, speed, and endurance. At the end of the selection, only 400 kids are selected for the next stage of the training. Selected kids then take part in one of five training camps that last four days. Professional instructors lead the training camps, in which the kids are taught all the rules and regulations required to successfully participate as ball kids at the tournament. Only 240 kids are then officially selected to take part in the French Open as ball kids, along with 10 kids from overseas territories of France or foreigners. Kids that participated in previous editions of the French Open can register through a cover letter and be selected for the new edition of the tournament. Thirty kids can be selected through this process, and the total number of official ball kids is 280 for each French Open edition.

A sample was made up of ball kids participating at the 2022 French Open at Roland Garros. From 22 May to 5 June 2022, the tournament was held in Paris, France. The total number of ball kids that participated in this tournament was 280, but for the purpose of this study, 26 were analyzed during several rotations of ball kids’ activity on the three main courts with different durations. Of those 26 ball kids, 20 were male and 6 female (N = 26, M = 20, F = 6; age = 15.00 ± 0.84; height = 169.03 ± 9.62; weight = 52.26 ± 7.35), as described in Table 1. Each ball kid took part in several rotations, which were then analyzed (data entry N = 94; number of rotations mean = 3.62 and 1.55). At the time of the study, the ball kids participated in sports activities 11.13 ± 3.92 h per week. All the participants were informed of the purpose, benefits, and risks of the investigation. The participants provided written consent signed by their parents.

### 2.2. Instruments

The ball kids that participated in this research answered a short questionnaire that consisted of two parts. The first part included descriptive information about the participant’s chronological age and a self-estimate of body height and weight. In the second part, participants assessed how many hours of sports they play per week and how many hours of tennis they play per week on average. Before and after each rotation of the ball kid’s activity on court (during match play), the participant’s rating of perceived exertion was measured using a standardized scale of 1–10, where 1 represented minimum effort and 10 represented maximum effort activity. Workload data was collected using a GPS device that collected position data for each movement [8]. The GPS device with Catapult GPS technology (Vector S7, Catapult, Catapult Sports Ltd., Melbourne, Australia) collected information about total distance covered (in meters), velocity of movement, and number of accelerations and decelerations. GPS collected information about the specific speed at which participants ran; walking and jogging were specified as speeds under 14.3 km/h, and running was specified as speeds above 14.3 km/h. Each participant wore a GPS device of the same brand to reduce measurement error when comparing results between participants, as some researchers suggested [9].

### 2.3. Statistical Analysis

Data for all rotations was analyzed together, which was N = 94 data entries (from 26 participants). The data was reported as means and standard deviations. Variables were assessed with the Shapiro-Wilk W normality test. Because the results showed a normal distribution, parametric statistics can be used for analysis. The variables that were analyzed were meters per minute of walking and jogging on court (*p* = 0.41), meters per minute covered on court (*p* = 0.09), and maximum velocity achieved on court (km/h; *p* = 0.67). The variable total number of decelerations per minute on court (*p* = 0.01) is not normally distributed; therefore, a nonparametric statistic should be used. A T-test for independent samples was used for analyzing differences between groups of ball kids that are positioned on the net or at the back of the court. Because each ball kid’s rotation lasted a different amount of time, the data was standardized by dividing the variable by the total time spent on court. All statistical analysis was performed using IBM SPSS Statistics version 22, and the level of statistical significance was established at *p* < 0.05. 

## 3. Results

Selected variables were analyzed, and their valid N, mean, minimum and maximum values, and standard deviation are shown in Table 1. The mean value of chronological age for all participants is 14.96 ± 0.84, which represents a homogenous sample. On average, all participants spend 11.13 ± 3.92 h per week participating in different sports activities and 5.84 ± 4.25 h per week playing tennis. This shows that all participants have a physically active lifestyle, and some are competitive athletes. 

The average rating of perceived exertion before a match rotation is 2.94 ± 1.30, when looking at all participants. All ball kids were well rested before most of their rotations. Rating of perceived exertion after the rotation is higher with an average of 4.55 ± 1.48.

Individual’s activity on court per rotation is an average of 41.38 ± 11.89 min, as shown in Table 2. During those rotations ball kids averaged 876.57 ± 310.94 m covered on court. Maximum velocity achieved on court by ball kids is 19.53 ± 1.78 km/h. To get a better perspective, these variables need to be analyzed by time spent on court. The average value of meters covered per minute spent on court is 21.42 ± 5.51. Walking and jogging represent the main activity on court (Mean = 18.81 ± 4.27). Figure 2 is a visual representation of the distribution of acceleration and deceleration intensities during the ball kid’s activity on court.

During the data analysis, a difference in workload between the two positions was noticed (Table 3). Selected variables were compared using a *t*-test for independent samples between groups: ball kids at the back of the court and by the net. Result of the statistical analysis showed a statistically significant difference between the two groups in variables meters covered per minute on court (t = 6.85, *p* = 0.00), total number of decelerations per minute (t = 8.39, *p* = 0.00), walking and jogging meters (<14.3 km/h) per minute (t = 4.68, *p* = 0.00) and maximum velocity achieved (t = 3.02, *p* = 0.00). A visualization of the differences between the groups of ball kids at the back of the court and by the net is presented in Figure 3.

It was also important to analyze the total workload of each ball kid during the research. Table 4 represents the mean values of every participant’s total workload. On average, each participant was engaged in 3.62 ± 1.55 rotations, ranging from only 1 rotation up to 5 rotations in total. Each participant spent an average of 146.46 ± 64.92 min on court and covered 3090.62 ± 1400.88 m. Ball kids completed 197.04 ± 99.71 number of accelerations (>0.5 m/s^2^) and 198.04 ± 99.32 of decelerations. 

## 4. Discussion

The purpose of this study was to examine the physical demands placed on ball boys during Roland Garros 2022. There are statistically significant differences between the ball kids at the net and in the back of the court in variables such as meters covered per minute on court, total number of decelerations per minute, walking and jogging meters per minute, and maximum velocity achieved. Such results have both scientific and practical applicability. As previously described, ball kids are individuals who retrieve and supply balls for players in professional tennis tournaments, and their activity helps speed up play and reduce inactive time. Tennis players react to information gathered through visual cues on court: their opponent’s position, racquet movement, ball height, and speed [10]. Similarly to them, ball kids react to the ball’s movement, height, and speed. Studies of young tennis players show that they cover 2.7–3.4 km and engage in high-intensity activity for 10–25% of the total distance covered [3,10]. Researchers indicate match characteristics such as rally duration (~8 s), effective playing time (~22%), and resting time between rallies (~18 s) [11,12]. Ball kids are inactive during rallies and the duration of the point. Ball kids are active during the resting time between points and rallies, and since resting time is significant in tennis, the ball kid’s activity is significant. Results in this study show that ball kids’ activity during matches is notable. On average, ball kids spend 41.38 ± 11.89 min on court and cover 876.57 ± 310.94 m. They make 55.96 ± 25.94 accelerations on average during a match, depending on their position and time on court. 

Ball kids participating at the French Open are young French tennis players aged 14.96 ± 0.84, as shown in Table 1. To the author’s knowledge, no research was conducted on ball kids; however, many studies analyzed young French tennis players. 

Study on eleven competitive French male tennis players (age: 13.4 ± 1.3) analyzed their physiological and performance outcomes when performing playing and nonplaying aerobic training. After the completion of the HIIT session, the rating of perceived exertion for the playing session was 8.45 ± 0.7 and for the playing session, 7.67 ± 0.7 [13]. Table 1 Shows that ball kids in this study had an average rating of perceived exertion of 4.55 ± 1.48 when finishing their rotation in the French Open. Considering that their activity was mainly aerobic with little high-intensity activity, that level of RPE is significant. Ball kids that participate in these kinds of events need to be physically and mentally prepared to endure the demands. The French Open is a major tennis tournament held over two weeks. This is a very limited time period with a congested match schedule for the young children in a very sensitive maturation period. There is a huge physical requirement for being a ball kid for the duration of a tournament. This research highlights the workload of ball kids during the French Open 2022. To perform well during the tournament’s two weeks, ball kids must be physically prepared in terms of lower body strength and mobility, power and speed, repeated sprint ability, and aerobic capacity. Furthermore, catching, picking up, and throwing balls requires optimal eye-hand coordination, which needs to be developed during the process. A training program targeting the main qualities and progressive overload has already been introduced for the 2022 edition and will be further developed for the 2023 edition of the tournament.

The high physical demands compared with normal settings may result in higher injury rates as well as mental distress. Thus, having all young ball kids available to perform during this important international competition is important to align injury risk factors (e.g., accumulated fatigue, reduced recovery time, and match load). According to Moreno-Perez et al. [14], during a season of high-performance junior tennis players (age = 17.2 ± 1.1), 3.5 injuries occur per 1000 h of tennis practice. Most of the injuries take place during the competition, with an injury rate of 23.8 injuries per 1000 h. In training, the injury rate is much lower, 3.4 injuries per 1000 h [15]. For the participants of this study, the ball kids, competition in this context represents activity during official tournament matches. Ball kids spent on average 146.46 ± 64.92 min on court where they covered 3090.62 ± 1400.88 m depending on the number of rotations they were included in. Participating as a ball kid at an event like this could help young athletes with building self-esteem, improving their social skills, and further developing their motor skills. The ball kids’ activity during a major international event can provide physical and psychological health benefits to these young athletes and specific sport motivation for their future sport involvement. 

### Limitations

This study has some limitations, which will be discussed below. Firstly, the participants involved in this research were selected as young ball kids in a very sensitive developmental period. Secondly, we were not able to analyze their biological status, which is very important at this stage; and thirdly, we did not have the possibility to observe their mental and physical fatigue that may have occurred during the testing period.

## 5. Conclusions

In summary, the present study suggests that there is a statistically significant difference between the ball kids at the net and in the back of the court in variables such as meters covered per minute on court, total number of decelerations per minute, walking and jogging meters per minute, and maximum velocity achieved. Similar investigations should be conducted in the future to expand our knowledge about this field and to improve and standardize testing protocols and selection criteria for potential ball kids. Creating an optimal test to improve the selection of ball kids should also be researched further.

## Figures and Tables

**Figure 1 ijerph-20-03793-f001:**
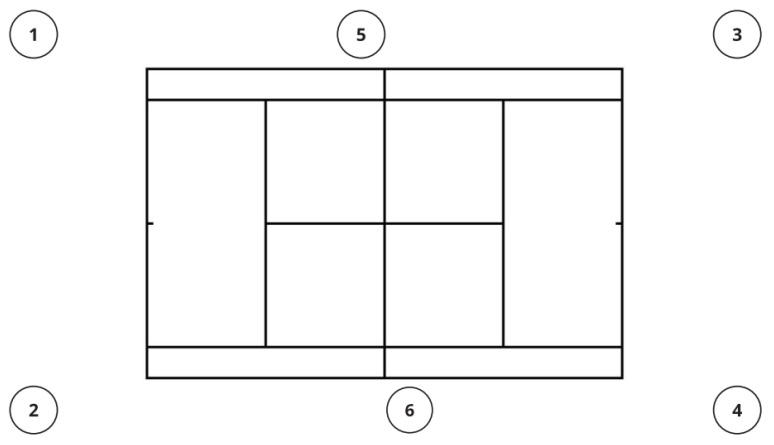
A visual representation of the ball kid’s court positions, with positions 1, 2, 3, and 4 representing back court positions and positions 5 and 6 representing net positions.

**Figure 2 ijerph-20-03793-f002:**
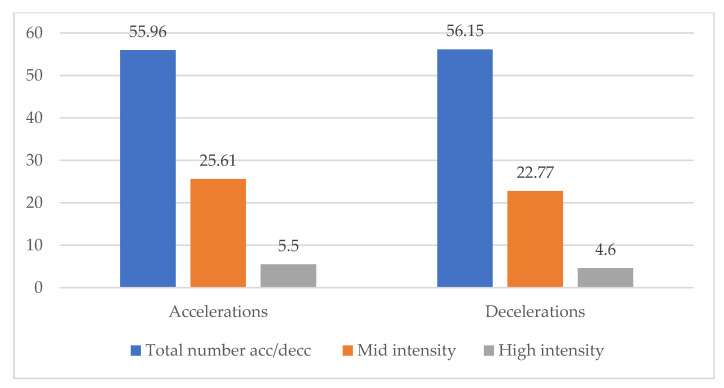
Visualization of acceleration and deceleration intensity distribution.

**Figure 3 ijerph-20-03793-f003:**
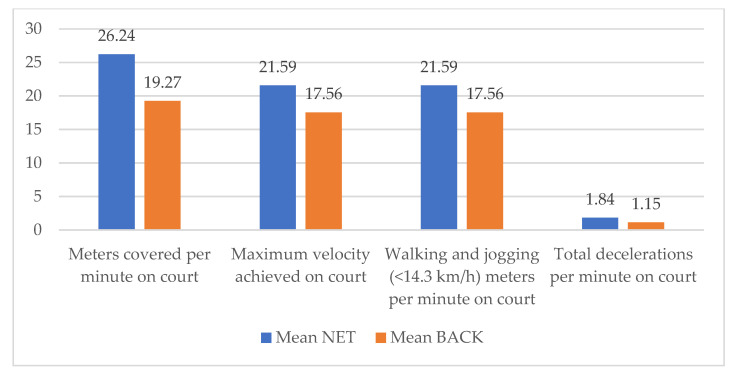
Visualization of differences between groups of ball kids at the net and at the back of the court.

**Table 1 ijerph-20-03793-t001:** Descriptive statistics of ball kids participating at the French Open, Roland Garros 2022.

Variables	Gender	Valid N	Mean	Minimum	Maximum	Std. Dev.
Chronological age	M	20	15.20	13.00	16.00	0.89
	F	6	14.50	14.00	15.00	0.55
Body height	M	20	169.60	145.00	187.00	10.73
	F	6	167.17	160.00	174.00	4.49
Body weight	M	20	51.95	35.00	63.00	8.29
	F	6	53.33	48.00	56.00	2.80
Hours per week sport	M	20	10.00	6.00	17.50	3.22
	F	6	14.92	8.00	20.00	3.93
Hours per week tennis	M	20	4.48	0.00	10.00	2.85
	F	6	10.50	3.00	16.00	5.54

**Table 2 ijerph-20-03793-t002:** Descriptive statistics of Ball Kid’s on-court activity.

Variables	Valid N	Mean	Minimum	Maximum	Std. Dev.
Total accelerations (>0.5 m/s^2^)	94	55.96	10.00	126.00	25.94
Mid intensity accelerations (>2 m/s^2^)	94	25.61	3.00	81.00	15.94
High intensity accelerations (>3 m/s^2^)	94	5.50	0.00	17.00	3.94
Mid intensity decelerations	94	22.77	3.00	71.00	13.44
High intensity decelerations	94	4.60	0.00	24.00	4.76
Total decelerations	94	56.15	11.00	127.00	25.47
Activity on court (minutes)	94	41.38	12.12	60.23	11.89
Total distance covered (meters)	94	876.57	193.00	1644.00	310.94
Maximum velocity achieved (km/h)	94	19.53	14.70	23.50	1.78
Meters covered per minute spent on court	94	21.42	11.18	36.40	5.51
Total decelerations on court per minute	94	1.36	0.49	2.51	0.49
Walking and jogging on court (<14.3 km/h)—meters per minute	94	18.81	10.38	30.69	4.27
Running (meters) (14.4–19.7 km/h)	94	2.51	0.02	7.57	1.89
Rating of perceived exertion IN	94	2.94	1.00	6.00	1.30
Rating of perceived exertion OUT	94	4.55	1.00	8.00	1.48

**Table 3 ijerph-20-03793-t003:** Using a T-test for independent samples between groups of ball kids on net and back positions, the analyzed variables are meters covered per minute on court, total decelerations per minute on court, walking and jogging meters per minute (<14.3 km/h) and maximum achieved velocity on court.

Grouping by Position: NET-BACK	MEAN NET	MEAN BACK	*t*-Value	*p*	Valid N NET	Valid N BACK
Meters covered per minutes on court	26.24	19.27	6.95	0.00	29	65
Total decelerations per minute on court	1.84	1.15	8.39	0.00	29	65
Walking and jogging- meters per minute on court (<14.3 km/h)	21.59	17.56	4.68	0.00	29	65
Running (meters) (14.4–19.7 km/h)	4.50	1.62	9.59	0.00	29	65
Maximum velocity achieved on court	20.33	19.17	3.02	0.00	29	65

**Table 4 ijerph-20-03793-t004:** Total workload per participant.

Variables	Valid N	Mean	Mean	Minimum	Maximum	Std. Dev.
Total time spent on court (minutes)	26	146.46	146.46	21.97	206.80	64.92
Total distance covered (meters)	26	3090.62	3090.62	374.00	5020.00	1400.88
Total number of accelerations (>0.5 m/s^2^)	26	197.04	197.04	17.00	360.00	99.71
Total number of decelerations	26	198.04	198.04	17.00	372.00	99.32
Number of rotations	26	3.62	3.62	1.00	5.00	1.55

## Data Availability

Data available on request.

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
