# Peer review of "“New Balls Please”: Physical Load Imposed on Ball Boys during the Roland Garros 2022"

_ijerph, 2023, doi:10.3390/ijerph20053793_

Round 1

Reviewer 1 Report

The topic of this research is original and may lead to future studies but the paper needs to be improved to make it easier to read.

Introduction :

- insert a figure showing the position of the ball boys

- paragraph 4 (line 67-80) should be before paragraph 3 (54-66)

Materials and Methods :

Participants :

- the distribution of girls and boys within the group of 26 is missing

- table 1 should be referenced in the text and corrected to include girls and boys

Results :

- the contents of tables 2 and 3 can be grouped in a single table

- delete the content that should go in the discussion section and the one already mentionned in the instruments section

- line 161 should be placed in the statistical analysis section

- figures 1 and 2 should be referenced in the text

- delete the excess column in table 5

Discussion :

- delete the first paragraph about tennis. First, try to summarize what the ball boys' activity is, and only then try to compare it to other activities already studied 

- focus the discussion on the physical requirements of being a ball boy for the duration of a tournament and make suggestions on training (and perhaps selection tests) that could be the topic of future research.

Author Response

Dear Editor:

We are pleased to resubmit for publication the revised version of Manuscript entitled "NEW BALLS PLEASE: PHYSICAL LOAD IMPOSED ON BALL BOYS DURING THE ROLAND GARROS 2022". We appreciate the constructive criticisms of the Editor and the reviewers. We have addressed each of their concerns as outlined below.

 Responses to the comments from Reviewer 1:

The topic of this research is original and may lead to future studies but the paper needs to be improved to make it easier to read.

Thank you this comment. We really appreciate your constructive criticisms. Thank you for pointing this out.

Introduction:

- insert a figure showing the position of the ball boys

A figure showing the position of the ball kids is inserted and added to the Methods section-participants because the reviewer 2 suggested that part to be moved there.

- paragraph 4 (line 67-80) should be before paragraph 3 (54-66)

Paragraph 4 is moved and placed before paragraph 3 as suggested.

Materials and Methods:

Participants:

- the distribution of girls and boys within the group of 26 is missing

Thank you this comment. The explanation of distribution of boys and girls within the groups of 26 was added.

- table 1 should be referenced in the text and corrected to include girls and boys

Table 1 was referenced in the text and corrected to include girls and boys and their descriptive data.

Results:

- the contents of tables 2 and 3 can be grouped in a single table

Tables 2 and 3 are now grouped as a single table named - Table 2. Descriptive statistics of analyzed Ball kids activity on court.

- delete the content that should go in the discussion section and the one already mentioned in the instruments section

Content about RPE scale and GPS tracking that was already mentioned in the instruments section was deleted.

- line 161 should be placed in the statistical analysis section

Line 161 was placed in the statistical analysis section as suggested.

- figures 1 and 2 should be referenced in the text

Two sentences referencing figures 1 and 2 (that are now called 2 and 3) were added.

- delete the excess column in table 5

Excess column was deleted.

Discussion:

- delete the first paragraph about tennis.

The paragraph about tennis and its references in the literature were deleted.

First, try to summarize what the ball boys' activity is, and only then try to compare it to other activities already studied 

Done.

- focus the discussion on the physical requirements of being a ball boy for the duration of a tournament and make suggestions on training (and perhaps selection tests) that could be the topic of future research.

Done.

We thank the editor and the reviewers again for their helpful comments, which we feel have improved our manuscript. We hope that with these modifications, our paper can now be accepted for publication.

Sincerely,

Dario Novak

Reviewer 2 Report

I have read the submitted article with great interest and I do find it very interesting since it shed light on an unexplored topic, ball kids.

However, in my opinion there are some things I would like to propose that hopefully can improve the article and its message.

General comments:

My biggest concern is not the research question within itself, it is more the structure of the paper. I believe the introduction; research question and discussion are not following each other or at least can be improved a lot. I would recommend staying with title PHYSICAL and see this article as a first step describing the load on the ball kids. The next step may be an intervention study or a study that brings other aspects into play, such as nutrition, mental health, injuries or others. So, my strong recommendation is to be clearer and do not grasp for to much. If you succeed with that, and I´m confident you will, I would very much like to see this paper published.

Title

The title is perfect, but let’s keep the main word PHYSICAL load in mind as we go along.

Introduction

Line 30-52 is describing the scenery of ball kids and their task, however, there are no references except for line 52. Furthermore, regarding the title PHYSICAL the paragraph gets to long before it targets the physical aspects. I would consider reorganizing the introduction (maybe move some of the sentences to methods ex; line 42 and forward). I understand this a study that have not been done before and therefore it is not so easy with references, but I think the information given in Line 54-66 Is good and should move up in the introduction since you are exploring PHYSICAL load on ball kids.

In line 67-76 you speculate about if the information explored can impact injury risk and/or act as injury prevention. I think it is better to stay with the main aim of your study (and title), PHYSICAL load. If you are about to explore injury risk, the design as you know needs to be prospective and although this can be a first step, I recommend that you do not involve injuries more than in one sentence as you did in line 75.

In line 78 you are covering the aim, so I do not think you need to finish off with the information in line 82 and 83.

Materials and Methods

I would consider moving some of the information in the introduction to this part (for example the “set up” of the ball kids).

Also, I would recommend referring to Table 1 “Descriptive” describing the study persons as you do in line 99 and forward to improve reading and understanding.

In line 116-119 (reference 6 and 7) may better be placed in introduction.

Statistics

I believe the statistics are straight forward and easy to understand, however, it is not my strongest area, so I leave that for other potential reviewers. Also, in this headline there are information that may be better suited in the methods (for example line 180-181).

Discussion

The first paragraph of the discussion I believe should repeat your main findings, most importantly you should not bring any new information/references into the article. I believe line 211-221 is information for the introduction and/or skip.

Line 222-223 is not the aim of the study, if you want to compare you should do this throughout your article, I recommend deleting this.

Once again in line 234-236 new information comes into play (ref 17). Furthermore (line 237-247), another twist is described in the same paragraph discussing nutrition, mental stress, intrinsic, extrinsic motivation and new references (18,19). I would recommend to deleting this since it is not the scope of your current study.

I believe the same goes for line 254-260 discussing COVID-19 and again new references.

If you are going to discuss injuries (not sure it is the aim) then you need to do this in the introduction (for example reference 24 with Moreno-Perez) but this also not the aim with your current study.

In my opinion the same goes for the mental health described in line 281 and forward.

Conclusion

Line 301-304 is perfect, summarizing the scope of your study. I think the rest of the conclusion is not part of your aims and therefore should not be concluded because you did not study that.

Author Response

Dear Editor:

We are pleased to resubmit for publication the revised version of Manuscript entitled "NEW BALLS PLEASE: PHYSICAL LOAD IMPOSED ON BALL BOYS DURING THE ROLAND GARROS 2022". We appreciate the constructive criticisms of the Editor and the reviewers. We have addressed each of their concerns as outlined below.

 Responses to the comments from Reviewer 2:

I have read the submitted article with great interest and I do find it very interesting since it shed light on an unexplored topic, ball kids. However, in my opinion there are some things I would like to propose that hopefully can improve the article and its message.

Thank you this comment.

General comments:

My biggest concern is not the research question within itself, it is more the structure of the paper. I believe the introduction; research question and discussion are not following each other or at least can be improved a lot. I would recommend staying with title PHYSICAL and see this article as a first step describing the load on the ball kids. The next step may be an intervention study or a study that brings other aspects into play, such as nutrition, mental health, injuries or others. So, my strong recommendation is to be clearer and do not grasp for to much. If you succeed with that, and I´m confident you will, I would very much like to see this paper published.

Thank you this comment. We really appreciate your constructive criticisms. Thank you for pointing this out.

Title

The title is perfect, but let’s keep the main word PHYSICAL load in mind as we go along.

Thank you this comment. We got the point!

Introduction

Line 30-52 is describing the scenery of ball kids and their task, however, there are no references except for line 52. Furthermore, regarding the title PHYSICAL the paragraph gets to long before it targets the physical aspects. I would consider reorganizing the introduction (maybe move some of the sentences to methods ex; line 42 and forward). I understand this a study that have not been done before and therefore it is not so easy with references, but I think the information given in Line 54-66 Is good and should move up in the introduction since you are exploring PHYSICAL load on ball kids. 

As suggested the lines 42-52 were moved to the methods section-participants together with a new figure showing the positions of ball kids like the reviewer 1 suggested. Lines 52-66 were also moved up as instructed.

In line 67-76 you speculate about if the information explored can impact injury risk and/or act as injury prevention. I think it is better to stay with the main aim of your study (and title), PHYSICAL load. If you are about to explore injury risk, the design as you know needs to be prospective and although this can be a first step, I recommend that you do not involve injuries more than in one sentence as you did in line 75.

Lines 67-72 were deleted as suggested.

In line 78 you are covering the aim, so I do not think you need to finish off with the information in line 82 and 83.

Line 78 that is covering the aim was left to conclude the introduction and information in lines 82-83 was deleted.

Materials and Methods

I would consider moving some of the information in the introduction to this part (for example the “set up” of the ball kids). 

The information from introduction about the set up of ball kids was moved.

Also, I would recommend referring to Table 1 “Descriptive” describing the study persons as you do in line 99 and forward to improve reading and understanding. 

Table 1 was named as suggested “Table 1. Descriptive statistics of ball kids participating at the French Open, Roland Garros 2022”

In line 116-119 (reference 6 and 7) may better be placed in introduction.

References 6 and 7 were moved to introduction.

Statistics

I believe the statistics are straight forward and easy to understand, however, it is not my strongest area, so I leave that for other potential reviewers. Also, in this headline there are information that may be better suited in the methods (for example line 180-181).

Lines 180-181 were moved to the methods as suggested.

Discussion

The first paragraph of the discussion I believe should repeat your main findings, most importantly you should not bring any new information/references into the article. I believe line 211-221 is information for the introduction and/or skip. 

The first paragraph was deleted.

Line 222-223 is not the aim of the study, if you want to compare you should do this throughout your article, I recommend deleting this. 

Lines 222-223 were deleted.

Once again in line 234-236 new information comes into play (ref 17). Furthermore (line 237-247), another twist is described in the same paragraph discussing nutrition, mental stress, intrinsic, extrinsic motivation and new references (18,19). I would recommend to deleting this since it is not the scope of your current study. 

Lines 234-236 and 237-247 were removed as noted in the comment. Also refferenced 18 and 19 were deleted.

I believe the same goes for line 254-260 discussing COVID-19 and again new references. 

Lines 254-260 discussing COVID-19 were removed.

If you are going to discuss injuries (not sure it is the aim) then you need to do this in the introduction (for example reference 24 with Moreno-Perez) but this also not the aim with your current study.

Done.

In my opinion the same goes for the mental health described in line 281 and forward.

Mental health described in line 281 and forward were removed as suggested.

Conclusion

Line 301-304 is perfect, summarizing the scope of your study. I think the rest of the conclusion is not part of your aims and therefore should not be concluded because you did not study that.

As suggested lines 301-304 were left as conclusion and the rest was removed.

We thank the editor and the reviewers again for their helpful comments, which we feel have improved our manuscript. We hope that with these modifications, our paper can now be accepted for publication.

Sincerely,

Dario Novak
